# Analysis of LoRaWAN 1.0 and 1.1 Protocols Security Mechanisms

**DOI:** 10.3390/s22103717

**Published:** 2022-05-13

**Authors:** Slim Loukil, Lamia Chaari Fourati, Anand Nayyar, K.-W.-A. Chee

**Affiliations:** 1Higher Institute of Business Administration, University of Sfax, Sfax 3018, Tunisia; slimloukil@yahoo.fr; 2ISIMS & SM@RTS/CRNS (Laboratory of Signals, systeMs, aRtificial Intelligence and neTworkS), Sfax 3018, Tunisia; lamiachaari1@gmail.com; 3Graduate School, Faculty of Information Technology, Duy Tan University, Da Nang 550000, Vietnam; anandnayyar@duytan.edu.vn; 4School of Electronic and Electrical Engineering, Kyungpook National University, Daegu 41566, Korea; 5School of Electronics Engineering, College of IT Engineering, Kyungpook National University, Daegu 41566, Korea

**Keywords:** Internet of Things, LoRaWAN, secure communication protocols, compatibility scenarios

## Abstract

LoRaWAN is a low power wide area network (LPWAN) technology protocol introduced by the LoRa Alliance in 2015. It was designed for its namesake features: long range, low power, low data rate, and wide area networks. Over the years, several proposals on protocol specifications have addressed various challenges in LoRaWAN, focusing on its architecture and security issues. All of these specifications must coexist, giving rise to the compatibility issues impacting the sustainability of this technology. This paper studies the compatibility issues in LoRaWAN protocols. First, we detail the different protocol specifications already disclosed by the LoRa Alliance in two major versions, v1.0 and v1.1. This is done through presenting two scenarios where we discuss the communication and security mechanisms. In the first scenario, we describe how an end node (ED) and network server (NS) implementing LoRaWAN v1.0 generate session security keys and exchange messages for v1.0. In the second scenario, we describe how an ED v1.1 and an NS v1.1 communicate after generating security session keys. Next, we highlight the compatibility issues between the components implementing the two different LoRaWAN Specifications (mainly v1.0 and v1.1). Next, we present two new scenarios (scenarios 3 and 4) interchanging the ED and NS versions. In scenario three, we detail how an ED implementing LoRaWAN v1.1 communicates with an NS v1.0. Conversely, in scenario four, we explain how an ED v1.0 and an NS v1.1 communicate. In all these four scenarios, we highlight the concerns with security mechanism: show security session keys are generated and how integrity and confidentiality are guaranteed in LoRaWAN. At the end, we present a comparative table of these four compatibility scenarios.

## 1. Introduction

Sustainable and smart cities have garnered interest from communities, organization, governments, and researchers alike. One of the main building blocks for smart cities and their applications is Internet of Things (IoT) technology. In the context of smart cities, smart systems addressing building management, parking, environment, health, mobility, etc., smart objects such as sensors and devices connect to the Internet and offload sensing data to remote computers for processing. Ericsson mobility reported that there are 14.6 billions IoT connections as of August 2021 [1]; these are predicted to increase to 30.2 billion by the year 2027. Indeed, the IoT technologies enable billions of objects and devices to connect and communicate, generating zettabytes of data and information (big data) circulating the Internet.

Several communication technologies have emerged to deliver data from objects to remote computers. 4G/5G/6G wireless communication [2], LoRaWAN [3], Bluetooth [4,5], RFID [6,7], ZigBee [8], WIFI [9], NB-IoT [10], and SIGFOX [11], are a few. Each of these technologies or communication protocols has unique features addressing the requirements of range, channel capacity, energy consumption, link speed, type of link, public/private infrastructure, radio frequency, security, etc. In this paper, we focus on LoRaWAN technology for its long range and low power consumption in resource-constrained IoT devices/objects. LoRa uses the chirp spread spectrum (CSS) spread spectrum technique allowing low power consumption, low throughput, and long range over the ISM band (a license-free band) to carry data. LoRaWAN protocol is developed and maintained by the LoRa Alliance. This alliance gathers telecommunication companies, equipment manufacturers, system integrators, sensor manufacturers, entrepreneurial start-ups, and semi-conductor companies. Since 2015, the LoRa Alliance has disclosed several specifications for LoRaWAN protocol, such as v1.0, v1.0.1, v1.0.2, v1.1, and v1.0.3.

From the published versions, we distinguished two major architectures. Essentially, LoRaWAN v1.0, v1.0.1, v1.0.2, and v1.0.3 specifications follow the same underlying architecture, referred to in this paper as LoRaWAN v1.0.x architecture. We also refer to LoRaWAN v1.1 as LoRaWAN v1.1 architecture. It is imperative that for seamless operability, it is not enough to study each architecture independently. Rather, for reliability purposes, it is important to see how the security mechanisms are implemented in each and whether they can operate interchangeably.

Distributed edge computing, including fog computing, is a main paradigm introduced recently to LoRaWAN, and it has facilitated the implementation of security algorithms. Indeed, Jakub et al. [12] tried to include the fog computing paradigm in LoRaWAN. The fundamental idea of this paradigm is to provide data processing and storage closer to the end-devices to reach higher efficiency in cases of large amounts of data. In this context, three IoT network architectures schemes adopting fog computing are proposed by the authors. A comparison of the proposed architectures in terms of service time was conducted by simulating each architecture to select the optimal architecture. Fog computing brings many benefits to IoT domains by reducing latency, decreasing bandwidth, improving efficiency, etc. However, security issues should not be ignored. In [13,14], the authors highlighted the security and privacy issues of fog computing in IoT.

Since security is a major concern when using communication technologies, we defined the objectives of this paper to:(i)Provide a comprehensive overview of the two LoRaWAN architectures’ communication protocols with a focus on security aspects.(ii)Highlight security-related compatibility issues between the two LoRaWAN major versions, v1.0 and v1.1.(iii)Analyze the compatibility issues within the two architecture components by interchanging NS and ED versions (homogeneity and heterogeneity). Furthermore, we analyze how security session keys are generated and how integrity and confidentiality are guaranteed in LoRaWAN.

This paper is organized as follows. In Section 2, we present LoRaWAN terminologies and protocol version details, we detail the LoRaWAN v1.0.x and v1.1 architectures, and we detail the protocol messaging structure. Section 3, we present a review of the LoRaWAN protocols, security issues, vulnerabilities, and challenges. Section 4 presents the four scenarios detailing the interoperability of the NSs and EDs of the two architectures. The first two scenarios describe the homogeneous equipment compatibility with the ED v1.0 and NS v1.0 security session context generation (OTA and ABP Activation), confidentiality in messaging, and the integrity and authenticity of messages. Scenario two describes the ED v1.1 and NS v1.1 security session context generation details and communication messaging details. Next, the heterogeneous equipment compatibility with ED v1.1 and NS v1.0 secure communications is detailed in scenario three, which is followed by scenario four detailing ED v1.0’s operability with NS v1.1. Finally we present the conclusions and discussion in Section 6.

## 2. LoRaWAN

In this section, we introduce LoRaWAN: What is the LoRa Alliance? What is LoRa? What is LoRaWAN protocol? What versions exist? After that, we detail different architectures introduced by LoRaWAN Specifications. Finally, we detail all messages exchanged between different entities in LoRaWAN.

### 2.1. LoRaWAN-Alliance

The LoRa Alliance is a non-profit organization bringing together stakeholders in the field of IoT and LoRa technology. Indeed, it is an alliance between several organizations, such as telecommunications companies, equipment manufacturers, system integrators, sensor manufacturers, entrepreneurial start-ups, and individuals interested in new technologies. This alliance forms an eco-system whose objective is to promote LoRa technology as a communication technology in the IoT sector. To achieve this goal, the LoRaWAN protocol was created. Additionally, the LoRa Alliance has to develop, promote, and standardize LoRaWAN protocol.

### 2.2. LoRa Technology

A low power wide area network (LPWAN) is a network suitable for interconnecting large-area low-power devices. LoRa (Long Range) is one network of this type. It was developed by Semtech Inc in 2014 [15]. LoRa is a modulation/demodulation scheme [16] derived from chirp spread spectrum (CSS). This scheme is known for its low power consumption and its robustness against noise [17]. LoRa modulation technology uses the ISM band (Industrial, Scientific, and Medical) which is unlicensed. In fact, the frequencies in ISM bands are different in different regions of the world. The latest frequency plans [18] are listed in Table 1 with their common names.

### 2.3. LoRaWAN Protocol

LoRaWAN protocol is a media access control (MAC) layer protocol. It was developed by the LoRa Alliance for connecting wireless and battery-powered devices deployed in large areas (regional, national, or global areas). Wireless communication is performed by LoRa technology. This protocol ensures bi-directional communication, end-to-end security, and mobility services.

Since 2015, the LoRa Alliance has published many versions of the LoRaWAN protocol, such as:LoRaWAN version 1.0 (January 2015) [19]: The first approved version of LoRaWAN 1.0.LoRaWAN version 1.0.1 (February 2016) [20]: This minor revision clarified some points that were not clear in the previous version, added some frequency plans, and made some modifications to MAC commands.LoRaWAN version 1.0.2 (July 2016) [21]: This minor revision added more MAC commands, fixed some errors in the previous version, and moved the Section "Physical Layer" to a separate document.LoRaWAN version 1.1 (October 2017) [22]: This major revision presented a new architecture by adding a new server called Join Server (JS), provided many enhancements in the security mechanism, such as using two root keys instead of one to derive the session security keys, and adding many countermeasures to avoid some vulnerabilities reported in the previous versions.LoRaWAN version 1.0.3 (July 2018) [23]: This minor revision related to v1.0.2 reused the same specifications for the class B devices mentioned in LoRaWAN version 1.1, added some MAC commands for class A devices, and deprecated some others.LoRaWAN version 1.0.4 (October 2020) [24]: This minor revision related to v1.0.3 provided some clarification on matters such as FCnt usage and behaviors, ADR behavior, channel selection procedure during joining, and retransmission backoff. We noticed two major security-related changes in this release: DevNonce generation is incremental instead of random, and AppEUI and AppNonce were replaced with JoinEUI and JoinNonce.

### 2.4. LoRaWAN Limitations

The LoraWAN protocol is suitable for several IoT applications. Noura et al., in [25], listed some IoT applications that might use LoRaWAN technology, such as smart homes and buildings, smart cities, smart healthcare, smart agriculture systems, smart parking, and smart metering. Nevertheless, LoRaWAN has some limitations that prevent it from being used in all IoT applications. Several research studies have evaluated the performance of this technology. Adelanto et al. in [26] conducted an analysis of LoRaWAN to demystify its capabilities and limitations. They noticed that the duty-cycle regulations on the ISM band limit the number of messages per day that can be transmitted by nodes. Additionally, they suggested that IoT applications using LoRaWAN must limit the use of acknowledgment frames. Hence, according to the authors, LoRaWAN networks are not suitable for ultra-reliable services. Finally, Adelanto et al. concluded that LoRaWAN technology can be used for IoT-application such as smart lighting, smart parking, smart waste, and logistics tracking, but that it is not suitable for real-time monitoring or video surveillance. Additionally, Bankov et al., in [27], evaluated the limits of LoRaWAN channel access. A mathematical model of the transmission process was developed. Their evaluation was performed on a network composed of N nodes connected to a gateway. The network operated on three main channels at six data rates (SF = 12 to SF = 5). Such a network has a capacity of 0.1 51 byte frames per second according to the authors’ experimentation, which corresponds to network traffic generated by 5000 devices each sending two messages per day.

Another limitation that needs to be mentioned for LoRaWAN is the presence of some security vulnerabilities. Although the various LoRAWAN specifications have dealt with security and many improvements have been proposed, the LoRaWAN protocol has some vulnerabilities. Several works have assessed the security of LoRAWAN and detected some potential attacks, such as join procedure attacks [28,29,30], denial-of-service attacks [31], and replay attacks [32,33].

### 2.5. LoRaWAN Architectures

From all versions disclosed by LoRa Alliance, we distinguish the use of two different architectures. LoRaWAN versions 1.0, 1.0.1, 1.0.2, 1.0.3, and 1.0.4 share the same architecture. Version 1.1 uses a different architecture. In what follows, LoRaWAN v1.0.x refers to LoRaWAN Specifications v1.0, v1.0.1, v1.0.2, and v1.0.3, and LoRaWAN v1.1 refers to LoRaWAN Specification v1.1.

In this section, we detail the two architectures: LoRaWAN v1.0.x architecture and LoRaWAN v1.1 architecture.

#### 2.5.1. LoRaWAN v1.0.x Architecture

The end-device (ED), gateways (GWs), network server (NS), and application server (AS) are the major components in a LoRaWAN network, as specified in LoRaWAN v1.0.x [19,20,21,22]. The architecture is organized as a star-of-stars topology, as depicted in Figure 1. The NS is connected to GW via secured standard IP connections. The connections between EDs and one or more GWs are wireless using LoRa radio-frequency technology in a single hop way [19,21].

An ED is a sensor or an actuator. It may be mobile or immobile. The sensor’s main function is to capture information and broadcast it. This is called uplink communication. This communication may be received by one or more GWs. Each uplink communication is followed by two receive windows allowing network servers to send data. This communication is called the downlink. Three classes of ED are used in LoRaWAN, which are A, B, and C. All EDs must implement class A functionality. This class allows an ED to send an uplink at any time. However, the GW must wait for the opening of the two receive windows, opened by the ED after each uplink, to send its downlink data. Class B adds, besides the two receive windows aforementioned, scheduled receive windows at fixed intervals. A time-synchronized beacon is needed. EDs of class C are available for downlink traffic nearly all the time, except when they are transmitting uplink data.

A GW is a bridge between wireless and wired parts of a LoRaWAN network. It forwards uplink communication from EDs to the NS and conversely forwards downlink communication from the NS to EDs.

The NS is the center of the star topology. It has several functions, such as forwarding uplink communication to ASs, forwarding downlink communication coming from ASs to appropriate EDs, performing security checks, and controlling the MAC layer of the EDs.

The AS processes all the application-layer payloads sent by the associated EDs. It also offers the application-level service to the end-user. All application layer downlink payloads are generated by this server.

An ED can send or receive data only after being properly configured. In the LoRaWAN Specification there are two ways to configure and activate an ED on the network: Activation by Personalization (ABP) is a manual configuration way and Over-The-Air Activation (OTAA) is an automatic configuration way.

#### 2.5.2. LoRaWAN v1.1 Architecture

In LoRaWAN v1.1 architecture, the network server is split into three servers: home network server (hNS), serving network server (sNS), and forwarding network server (fNS) (see Figure 2). This architecture enables roaming services between network operators. In fact, the LoRaWAn Alliance defines two types of roaming: passive roaming and handover roaming [23].

When an ED is under the coverage of GWs owned by their home network, the hNS controls uplink and downlink packets along with the MAC layer of this ED. The hNS stores the device profile, service profile, routing profile, and DevEUI (unique identifier) of the ED. The hNS interfaces with the AS to deliver the application payload.

When an ED is covered by GWs belonging to another network server, the ED can perform handover roaming or passive roaming. In the first case, the control of the MAC layer and the uplink and downlink packets of that ED becomes the responsibility of this server. This server is called the serving network server (sNS). In the second case, the server of the visited network only takes packets between the ED and sNS. The control of the MAC layer of the ED remains the responsibility of the sNS. This server is called a forwarding server network (fNS). To support roaming signaling and payload delivery, fNS interfaces with sNS and sNS interfaces with hNS.

Another important enhancement in loRaWAN v1.1 is a new server called Join Server (Js). This server orchestrates the activation process for the Over-The-Air (OTA) ED instead of NS. The JS has the responsibility of deriving all session security keys. Additionally, it must deliver, in a secure way, the appropriate keys for AS and NS. The presence of this server brings trust between network components (NS and AS). The JS interfaces with hNS, sNS, and fNS.

### 2.6. Basic Messages with LoRaWAN

#### 2.6.1. Message Structures Used in LoRaWAN v1.0.x

In the physical layer, every message over LoRaWAN is comprised of a preamble, a physical header (PHDR), a physical header CRC (PHDR_CRC), and a physical payload (PHYPayload). For uplink messages, a CRC field is added.

The physical payload (PHYPayload) is formed by a MAC header specifying the message type and the MAC major version (up to now, all LoRaWAN versions share the same value of 0, indicating LoRaWAN release 1.x), a MAC payload, and a 4-byte message integrity code (MIC). These form the MAC layer of the LoRa packet (see Figure 3, red part). In LoRaWAN v1.0, six different message types can be exchanged:
Join-request: This type of message is used each time an end-device (ED) wants to join the network server (NS) in order to open a new session with it. This message is sent by an ED after powering on or resetting, or when the terminal wants to change its security session keys.Join-accept: This type of message is a response to the join-request message. It is created and sent by JS if the ED has permission to join the network.

These two messages are only used in the OTA activation mode. This activation mode is used to automatically generate a session context between an ED and an NS.

Once an ED has a session context via OTA (automatic) or ABP (manual) activation, it can exchange the following types of messages with the NS:Unconfirmed dataUp: this type of message is sent by an ED to an NS when the terminal decides to send data that do not require acknowledgment.Unconfirmed dataDown: this type of message is sent by an NS to an ED when the NS has data to send which does not require acknowledgment. It can only be sent during one of the two reception windows opened by an ED after an uplink communication.Confirmed dataUp: This type of message is distinguished from unconfirmed dataUP by the need for an acknowledgment from the receiver (NS).Confirmed dataDown: This type of message is distinguished from unconfirmed dataDown by the need for an acknowledgment from the receiver (ED).

For all of these message types except join-request and join-accept, the MAC payload contains a frame header field (FHDR) followed by two optional fields: port field (Fport) and frame payload (FRMPayload). The frame header is comprised of a device address (DevAddr), a frame control (FCtrl), a 2-byte frame counter (FCnt), and up to 15 bytes of frame options (FOpts) used to carry MAC commands (see Figure 3, green part). The maximum length of the MACPayload field is M bytes. Likewise, N is the maximum size of the FRMPayload field. M and N are region-specific values. For example, in Europe, M and N can reach, respectively, 250 bytes and 242 bytes.

As depicted in Figure 3, the MACPayload of the join-request message is comprised of AppEUI (unique identifier of an application), DevEUI (unique identifier of an ED), and DevNonce (random number). The purposes of each field are described in Section 4.1.1.

The fields in a join-accept message are depicted in Figure 3. The purposes of each one are described in Section 4.1.1.

#### 2.6.2. Message Structure Used in LoRaWAN v1.1 Architecture

In addition to the six types of messages used in LoRaWAN v1.0.x, a new type has been added called rejoin-request. This type of message is used for roaming purposes, and changing or restoring security session keys. There are three types of rejoin-request (types 0, 1, and 2).

In LoRaWAN v1.1, the messages have the same structure as that used in the previous version except for a single field for data-up messages: the bit field between ACK and FOptsLen located at the FCtrl field became ClassB. This field is used to inform the NS whether the device is in class B mode (when it is set) or not.

In a join-request message, the only change is that the JoinEUI field has replaced the AppEUI Field.

In a join-accept message, the JoinNonce field has replaced the AppNonce field. Additionally, in DLSettings, the RFU field is replaced by the OptNeg field. When this field is set to one, it indicates to ED that NS implements LoRaWAN v1.1; otherwise, it is unset.

For a rejoin-request message, LoRaWAN v1.1 distinguishes three types: 1, 2, and 3. The rejoin-request types 0 and 2 share the same structure, as illustrated in Figure 4. A rejoin-request type 1 has nearly the same fields as a join-request message, as depicted in Figure 4.

## 3. Related Works

The authors in [34] provided a taxonomic overview of the Industrial Internet of Things (IIoT) infrastructure and security requirements recommended by the standards bodies Industrial Consortium and OpenFog Consortium. According to the authors, most IoT deployments follow a three-tiered architecture: edge tier, platform tier, and enterprise tier. Considering this architecture, and for each tier, the authors analyzed and discussed different security protocols, trying to answer the following questions: how secure the connectivity is ensured to be, how the different protocols operate with each other, what the security vulnerabilities are, and how one can mitigate them.

Sundaram et al. [35] studied the research challenges of LoRa networking. A taxonomy was proposed to categorize them. The major challenges presented by the authors are energy consumption, communication range, multiple access, error correction, and security.

Lalle et al. [36,37] conducted a comparative study between LoRaWAN and the two main competitors, Sigfox and NB-IoT, for smart water grid applications. This study was carried out on the basis of six criteria: power consumption, latency, scalability, quality of service, cost, communication range, and coverage.

Kuntke et al. [38] inspected 37 articles in a systematic literature review to build a list of vulnerabilities for multiple LoRaWAN versions up to v1.0.4. Their study detected 16 attacks and classified them into six attack types: physical attacks, message replays, traffic analysis, denial of service, spoofing attacks, and other attacks. Additionally, the authors collected from the literature countermeasures for mitigation. Additionally, the authors evaluated the practicality of the collected vulnerabilities in the case of agriculture IoT.

De moraes et al. [39] conducted a systematic review of security in LoRaWAN v1.0 and v1.1. To perform this systematic review, the authors employed a methodology in accordance with guidelines established by Kitchenham. This methodology is divided into three phases: planning, conducting, and reporting. In this review, the authors identified 19 vulnerabilities in the LoRaWAN protocol. Additionally, the authors revealed that 77.5% of studies are interested in LoRaWAN v1.0 compared to 22.5% for LoRaWAN v1.1.

Hessel [40] identified the most relevant attacks against LoRaWAN by performing a literature review. After that, an attack catalog was created, and each attack was discussed and evaluated for the different LoRaWAN versions. The author provided a tool to assess the security of real LoRaWAN networks. An experimental evaluation of the attacks was performed and revealed that the jamming attack is a serious threat to the availability of LoRaWAN networks.

Claverie el al. [41] proposed a security assessment platform for LoRaWAN v1.0. This platform can be used as a benchmark tests to study and evaluate the radio layer of the protocol.

Naoui et al. in [12] described some LoRaWAN vulnerabilities in the activation mode (ABP or OTAA) of end nodes and proposed a new secure scheme to counter these vulnerabilities. Mainly, they suggested using a one time password (OTP) generator to generate new session keys from existing ones after each reset. Additionally, they proposed to use timestamps instead of nonces for the join-request and the join-accept messages to prevent old messages from being replayed. Additionally, they suggested entrusting the task of generating session security keys to the application server, considering it as a trusted entity.

Eldefrawy et al. [13] conducted a formal security analysis using the Scyther tool on LoRaWAN v1.0 and v1.1. They concluded that LoRaWAN attacks on non-injective agreement claims and non-injective synchronization attacks occur in LoRaWAN v1.0 and do not occur in LoRaWAN v1.1.

In [14], Butun et al. conducted a comprehensive security risk assessment. Firstly, they discussed some security challenges for LoRaWAN v1.1. After that, they created a threat catalog. Following ETSI guidelines [42], the authors analyzed each threat from the created catalog. Their analyses ended with three major threats: end-node physical capture, rogue gateways, and self-replay. Additionally, Butun et al. proposed some countermeasures to reduce these risks.

Butun et al. [43] discussed some security threats of LoRaWAN v1.1 based on an analysis of the protocol and formerly known security issues. The most relevant attacks presented in the paper are RF jamming attacks, replay attacks, beacon synchronization attacks, network traffic analysis, and man-in-the-middle attacks against servers. Additionally, they performed a security scan using the Scyther security verification tool. It appears from this analysis that LoRaWAN is cryptographically secure.

Haxhibeqiri et al. [44] surveyed different works evaluating the performances of the LoRa and the LoRaWAN. Additionally, they conducted strengths, weaknesses, opportunities, and threats (SWOT) analysis. According to their analysis, LoRaWAN has six strengths (cheap end-devices, private network deployment, etc.), five weaknesses (performance of ADR mechanisms in congested traffic, low scalability in downlink communication, etc.), four opportunities (further decreasing the power usage, avoiding duty cycling by CSMA schemes, etc.) and three threats (security issues, especially for LoRaWAN v1.0, interference with other technologies, etc.).

Yang et al. [45] presented five vulnerabilities in the LoRaWAN protocol. They also implemented all attack vectors in a hardware or software proof-of-concept. Firstly, they showed that eavesdropping of two packets encrypted with the same counter leads to decryption of the content, since packets are encrypted using AES in counter mode. Secondly, they indicated that an attacker can use old messages in a new session in case the security keys for that session have not been changed. This is more likely for devices activated by ABP. Thirdly, they stated that a man-in-the-middle attack can be performed between LPWAN operators and application providers. This attack is made possible due to the early termination of message integrity checking at the network server. Fourthly, they noticed that Ack message does not indicate which message is confirmed. Thus, an ACK spoofing attack is possible. The latest attack targeted end-devices of class B, in which beacon frames are broadcast without encryption. This vulnerability can lead to spoofing the location of the LoRa gateway or exhausting the battery. Yang et al. proposed some solutions to those vulnerabilities.

Jakub et al. [46] tried to include the fog computing paradigm in LoRaWAN. The fundamental idea of this paradigm is to provide data processing and storage closer to the end-devices to reach higher efficiency for large amounts of data. In this context, three IoT network architectures adopting fog computing were proposed by the authors. A comparison of the proposed architectures in terms of service time was conducted by simulating each architecture to select the optimal one. Fog computing brings many benefits to IoT domains by reducing latency, decreasing bandwidth, improving efficiency, etc. However, security issues should not be ignored. In [47,48], the authors highlighted the security and privacy issues of fog computing in IoT.

## 4. LoRaWAN Compatibility Scenarios: Basic Security Concepts

Both versions of LoRaWAN must co-exist, and this imposes compatibility issues. LoRaWAN v1.1 has addressed this problem by proposing alternatives. In fact, a network server implementing LoRaWAN v1.1 (NS v1.1) must communicate with an ED implementing LoRaWAN v1.1 (ED v1.1) and with an ED implementing LoRaWAN v1.0.x (ED v1.0). On the other hand, a network server implementing LoRaWAN v1.0.x (NS v1.0) must communicate with an ED v1.1 and with an ED v1.0.x. The network server has to decide which protocol version to be used. In fact, the NS selects the most common version between itself and the ED. In addition, it must communicate its decision to the ED.

In this work, we describe four scenarios of LoRaWAN. The first scenario describes the case of an ED v1.0 controlled by an NS v1.0. The second scenario describes the case of an ED v1.1 controlled by an NS v1.1. The third scenario addresses the case of an ED v1.1 controlled by an NS v1.0. The last scenario describes the case of an ED v1.0 controlled by an NS v1.1. For these four scenarios, we study only the case of class A devices. In the following sections, we will present each scenario in more detail with a focus on the security context: how security session context is generated and how the confidentiality, integrity, and authenticity of messages are ensured.

### 4.1. First Scenario: ED v1.0 with NS v1.0

In LoRaWAN v1.0.x, the security session context is generated manually in cases of ABP activation or automatically for the OTAA case. The security session context consists of:Sixteen-byte wide security session keys (NwkSkey and AppSkey),Two or three-byte frame counters (FCntUp and FCntDown),A four-byte address (DevAddr) to identify the ED in the network.

#### 4.1.1. Security Session Context Generation (OTA Activation)

Before activation, the ED must store in a secure way DevEUI, AppEUI, and AppKey. The DevEUI and the AppEUI are global identifiers in IEEE EUI64 address space that, respectively, identify the end-device and the application provider (ED’s owner). The AppKey is an AES-128 root key assigned to the ED during fabrication. It is used to derive all the security session context. This root key must be unique for each ED. Otherwise, compromising the Appkey of one ED affects all other EDs that use the same key.

The activation procedure in OTA mode is performed by exchanging join-request and join-accept messages between the ED and the NS, as depicted in Figure 5.

The join-request message is sent by the ED requesting the NS to create a new session context. It contains three pieces of information: AppEUI, DevEUI, and DevNonce. The DevNonce is a number (16 bits wide) randomly generated by the ED. For each ED, the NS must store a certain number of DevNonce used in the past and discard all joint-request messages with any of these stored DevNonce. In this way, replaying a recorded join-request message is not allowed. The join-request message is sent without being encrypted; however, it is signed. The details about how to sign this message are presented in Section 4.1.4.

If the ED has permission to join a network, the NS creates a joint-accept message and sends it. This message contains an AppNonce (unique ID of 3 bytes), a NetID (network ID of 3 bytes), a DevAddr (end-device address of 4 bytes), a DLSettings (downlink parameters of 1-byte), a RxDelay (delay between TX and RX windows of 1-byte), and a CFList (optionally a list of network parameters of 16 bytes). The AppNonce (24 bits wide) is a random value or some form of unique ID provided by the NS. The joint-accept message is signed and then encrypted. The details about how to cipher and sign this message are presented in Section 4.1.3 and Section 4.1.4.

To derive the security session keys (NwkSKey and AppSKey), the generic algorithm AES128_encrypt [49] described in the Institute of Electronics Engineers (IEEE) 802.15.4/2006 is used with a key-size of 128 bits (AppKey is a root key shared by the ED and the NS). Each session key is calculated as follows:(1)$$\begin{array}{c} \mathrm{NwkSKey}=\mathrm{aes}128\_\mathrm{encrypt}(\mathrm{AppKey},0x01|\mathrm{AppNonce}|\mathrm{NetID}|\mathrm{DevNonce}|{\mathrm{pad}}_{16})\\ \mathrm{AppSKey}=\mathrm{aes}128\_\mathrm{encrypt}(\mathrm{AppKey},0x02|\mathrm{AppNonce}|\mathrm{NetID}|\mathrm{DevNonce}|{\mathrm{pad}}_{16}) \end{array}$$

Once the activation procedure succeeds, the NS and ED will share the same security session keys. In addition, they must initialize the two frame counters (FCntUp and FCntDown) to zero. The FCntUp will be incremented by the ED on each uplink, and the NS must keep track of it. Conversely, the FCntDown will be incremented by the NS on each downlink, and the ED must keep track of it. The ED and NS must ignore all dataUp or dataDown with frame counter values less than the recorded one. This mechanism prevents replaying a previously recorded message due to a malicious device.

From now, EDs and NSs can exchange encrypted and signed messages as shown in Figure 6.

#### 4.1.2. Security Session Context Generation (ABP Activation)

With this means of activation, the ED is tied to a particular LoRaWAN network. Before starting the communication, all the security session keys (AppSKey and NwkSKey) and DevAddr are hard-coded into the device and stored on the NS by an out-of-band mechanism. The two frame counters (FcntUp and FCntDown) are initialized to zero after each reset. According to LoRaWAN v1.0.x [19,21], each ED should have a unique set of security session keys. In this way, compromising the keys of one device does not affect others. In ABP activation mode, The NS must store the AppEUI for each ED with an out-of-band mechanism. The dataup and the dataDown are exchanged between the ABP end-device and the network server, as illustrated in Figure 6.

#### 4.1.3. Confidentiality of Messages

The join-request message is sent by ED without being encrypted. Thus, every device with the capability of intercepting this message can read it.

In the join-accept message, the fields encrypted by NS are AppNonce, NetID, DevAddr, DLSettings, RxDelay, CFList, and MIC (details of calculating the MIC field are presented in Section 4.1.4). The AppKey with aes128_decryp is used for encryption as follows:(2)aes128_decrypt(AppKey,AppNonce∣NetID∣DevAddr∣DLSettings∣RxDelay∣CFList∣MIC)

Since the NS uses the aes128_decrypt function to encrypt the message (AES decrypt operation in ECB mode), the ED must use aes128_encrypt to decrypt it. In this way, ED only has to implement the function aes128_encrypt for the whole process of encryption.

In LoRaWAN v1.0.x, all the fields in a dataUp/Down message are sent in clear except FRMPayload. In fact, this field contains the data application sent by AS or MAC commands sent by NS according to the FPort field value (0 for MAC commands or [1…223] for application data). If FRMPayload carries MAC commands, the NwkSeky is used for encryption; otherwise, the AppSKey is used.

The encryption of the FRMPayload field is done using AES in a counter (CTR) mode, as depicted in Figure 7. The CTR mode consists of generating a stream of keys that will be XOR-ed with FRMPayload. The following steps describe how encryption is done:aDefining a sequence of blocks $A_{i}$/i = 1 k (k = ceil(len(FRMPayload)/16)) that mainly contains Dir (0 for dataUp or 1 for dataDown), DevAddr, a counter (FCntUp for dataUp or FCntDown for dataDown), and i. Each block has a size of 128 bits.bCalculating a stream of keys (a sequence S of blocks $S_{i}$) by encryption the sequence of blocks $A_{i}$, already calculated above, using the AES algorithm.
S = $S_{1}|S_{i}|\ldots |S_{k}$$S_{i}$ = aes128_encrypt(K,$A_{i}$) i = 1 kK = NwkSKey or AppSkeycThe encryption or decryption is calculated by truncating (FRMPayload$|{\mathrm{pad}}_{16}$) XOR S to the first len(FRMPayload) bytes.

#### 4.1.4. Integrity and Authenticity of Messages

All messages exchanged between ED and NS are signed by the use of the message integrity code (MIC). Its length is four bytes. It is calculated using the aes128_cmac function [50]. The MIC is calculated over all the fields in the message.

Actually, the MICs of joint-request, join-accept, dataUp, and dataDown are not calculated using the same security key. Suppose X designates all the fields of the message to be sent.

For a join-request/join-accept message, the MIC is calculated as follows:cmac = aes128_cmac(AppKey, X)MIC = cmac[0…3]

For a dataUp/Down message, the MIC is calculated as follows:Defining block $B_{0}$ which is mainly composed Dir (0: dataUp, 1: dataDown), DevAddr, FCntUp or FCntDown and len(X).cmac = aes128_cmac(NwkSKey, $B_{0}|$X)MIC = cmac[0…3]

### 4.2. Second Scenario: ED v1.1 with NS v1.1

#### 4.2.1. LoRWAN Enhancements in v1.1

LoRaWAN v1.1 introduced a roaming architecture. Thus, there are three different servers (hNS: home network server, sNs: serving network sever, and fNS: forwarding network server), instead of one server in the previous version (v1.0.x), with different roles:fNS: only forward messages exchanged between ED and sNS or EDs and hNS.sNS: forwards messages between hNS and EDs directly or via fNS and sends MAC commands to EDs directly or via fNShNS: forwards messages between EDs and the application server (AS) directly or via sNS or fNS, and sends MAC commands to ED directly or via sNS or fNS

The presence of three servers imposes strategic changes for security: the number of keys used, the way in which the security session keys are generated, the encrypting, and the signing of messages. Thus, two root keys were added (AppKey and NwkKey) instead of one. The AppKey is the root key used to perform the security of application data. The NwkKey is the root key used by the network operator to perform the security of MAC Command and to check the authenticity of messages. The use of these two root keys instead of one ensures the notion of trust between the AS and the NS.

From these two root keys, four security session keys are derived:AppSKey: derived from AppKey and used for data encryption,NwkSEncKey: derived from NwkKey used for MAC command encryption,SNwkSIntKey: derived from NwkKey used by hNS or SNs for checking message integrity,FNwkSIntKey: derived from NwkKey used by hNS or FNs for checking message integrity.

For OTA activation, LoRaWAN v1.1 added a new server called Join Server (JS). This server handles requests from ED to join the network. Moreover, it generates all the session security keys. The addition of this server (JS) improves the separation of trust between network servers and the application server. In this version, the ED must be attached to a specific join server by storing its identifier called JoinEUI, a global application ID in IEEE EUI64 address space. Two specific lifetime keys (JSEncKey and JSIntKey) are used to encrypt and authenticate join-accept answers when an ED sends a rejoin-request message.

All the security session keys are calculated from a root key (AppKey or NwkKey), JoinNonce, joinEUI, and DevNonce. The JoinNonce replaced AppNonce in the previous version of LoRaWAN. The DevNonce (16 bits wide) and JoinNonce (24 bits wide) are two device-specific sequential numbers starting at 0 and are incremented with each join-request/join-accept message.

As another enhancement in this newer version, all frame counters in LoRaWAN v1.1 must be 32 bits wide. Additionally, the frame counter for the dataDown has been split into two counters: NFcntDown and AFCntDown. The first counter is used for MAC commands and managed by a network server. The second one is used and managed by an application server.

#### 4.2.2. Security Session Context Generation: OTA Activation

Before activation, the ED must store in a secure way: DevEUI, JoinEUI (instead of AppEUI), AppKey, NwkKey, and the counters (DevNonce, Last_JoinNonce, RJcount0, RJcount1). The NS must store for each end-device: DevEUI, MAC Version, Last_DevNonce, Last_RJcount0, and Last_RJcount1. The JS must store for each ED: DevEUI, AppEUI, NwkKey, AppKey, and JoinNonce.

Keeping track of JoinNonce and DevNonce counters prevents replaying join-request and join-accept messages. Additionally, keeping track of RJcount0 and RJcount1 prevents replaying all types of rejoin-request messages. RJcount0 counter is incremented each time the end-device sends a rejoin request of type 0 or 2. RJcount1 counters are incremented each time the ED sends a rejoin request of type 1.

The two root keys (AppKey and NwkKey) are only stored in the ED and JS. In this way, JS plays the role of a third party, guaranteeing the trust.

In LoRaWAN v1.1, the activation of an end-device can be at home (end-device is under the coverage of home network server) or by roaming (end-device under the coverage of visited network server). In the last case, the activation may be passive roaming or handover roaming.

For the first time or after resetting, an OTA ED must begin its communication by sending a join-request message. During an ongoing open session, an OTA ED can change its session context by sending a rejoin-request message. Thus, an OTA-ED can have four different states depending on the server that is under its coverage area and the agreement with that server. All the states that ED can have are depicted in Figure 8. Roaming procedures and different scenarios of activation are described in the LoRaWAN Backend Interfaces 1.0 Specification [51].

A.Activation a new session for the first time

The ED sends a join-request to the server under its range. If the server is the home server (hNS), it forwards this request to the join server (this case is illustrated in Figure 9). Otherwise, it sends an HRStartReq message carrying the join-request to the hNS as a request for starting handover roaming. In this case, the hNS decides to enable handover or passive roaming according to the agreement between the two servers (home server and visited server) and sends to the network server an HRStartAns carrying the roaming activation type. When passive roaming, the visited server is called the forwarding network server (fNS). When handover roaming, the visited server is called the serving network server (sNS). These three cases are illustrated by green arrows in Figure 8.

The join-request message contains MHDR, JoinEUI, DevEUI, DevNonce, and MIC. This request is sent without being encryption; however, it is signed with NwkKey (details about signing this message are presented in Section 4.2.5). If the join-request is received by the hNS, it forwards it to the server1. This server checks the authenticity of the request (verifying MIC value) and generates an incremental number JoinNonce (3 bytes wide). After that, the JS creates a join-accept message and sends it to the hNS (see Figure 9). All the fields of this message are the same as in the previous version except for two things. The first is that AppNonce is replaced by JoinNonce. The second is the use of OptNeg bit in the DLSettings field. This bit is set to 1, indicating that the server implements LoRaWAN v1.1. Otherwise, this bit is set to 0.

If the home server network (hNS) is also the serving network, it forwards the received message to the ED. Otherwise, the hNS sends the join-accept message to the sNS to be forwarded to the ED later (this case is not studied here).

Now, the JS and the ED can derive the security session keys as follows:(3)$$\begin{array}{c} \mathrm{AppSKey}=\mathrm{aes}128\_\mathrm{encrypt}(\mathrm{AppKey},0x02|\mathrm{JoinNonce}|\mathrm{JoinEUI}|\mathrm{DevNonce}|{\mathrm{pad}}_{16})\\ \mathrm{FNwkSIntKey}=\mathrm{aes}128\_\mathrm{encrypt}(\mathrm{NwkKey},0x01|\mathrm{JoinNonce}|\mathrm{JoinEUI}|\mathrm{DevNonce}|{\mathrm{pad}}_{16})\\ \mathrm{SNwkSIntKey}=\mathrm{aes}128\_\mathrm{encrypt}(\mathrm{NwkKey},0x03|\mathrm{JoinNonce}|\mathrm{JoinEUI}|\mathrm{DevNonce}|{\mathrm{pad}}_{16})\\ \mathrm{NwkSEncKey}=\mathrm{aes}128\_\mathrm{encrypt}(\mathrm{NwkKey},0x04|\mathrm{JoinNonce}|\mathrm{JoinEUI}|\mathrm{DevNonce}|{\mathrm{pad}}_{16}) \end{array}$$

Besides these security session keys, the JS and ED use two lifetime keys: JSEncKey and JSIntKey. The first one is used for encryption a join-accept message if it is a reply to a rejoin-request. The second is used for signing rejoin-requests of type 1 and join-accept messages. These two keys are derived from NwkKey as follows:(4)$$\begin{array}{c} \mathrm{JSEncKey}=\mathrm{aes}128\_\mathrm{encrypt}(\mathrm{NwkKey},0x05|\mathrm{DevEUI}|{\mathrm{pad}}_{16})\\ \mathrm{JSIntKey}=\mathrm{aes}128\_\mathrm{encrypt}(\mathrm{NwkKey},0x06|\mathrm{DevEUI}|{\mathrm{pad}}_{16}) \end{array}$$

B.Reactivation During an Ongoing Session

In the case of a mobile ED, already activated, it is possible to switch from one operator to another by performing handover or passive roaming, as depicted in Figure 8 (orange arrows).

When a network server receives a dataUp message from an ED which it is not its serving network server (sNS or hNS), it sends a PRstartReq message to the sNS of this ED. This request initiates a passive roaming procedure. According to the roaming agreement between two operators, the sNS responds with a PRStartAns message indicating if the passive roaming is accepted or not. Once the result of PRstartReq is successful, the server becomes the forwarding server network (fNS), as illustrated in Figure 8. The mission of the fNS is to relay the messages exchanged between the hNS (or sNS) and the ED. The control of the ED via MAC commands is always the responsibility of the hNS (or sNS).

To perform handover roaming for an ongoing session, the ED must send a rejoin request of type 0. This request contains MHDR, Rejoin Type = 0, NetID (identifier of the device’s home network), DevEUI, and RJcount0 (2 bytes wide). The last part is a counter incremented with every rejoin request of type 0 or 2. The sNS must keep track of the last RJcount0. This prevents replaying rejoin requests of type 0 and 2. Once the end-device receives a valid join-accept, RJcount0 is reset to 0. If RJcount0 reaches its maximum value, the ED stops sending rejoin-request messages and goes back to the join state.

A rejoin request of type 0 can be sent by an ED periodically and autonomously or as a response to a ForceRejoinReq MAC command received from an sNS.

A rejoin request of type 2 is only used when the ED receives a ForceRejoinReq MAC command from the sNS. The server sends this command whenever it decides to refresh the session keys, reset frame counters, or change the DevAddr. Rejoin requests of types 0 and 2 have the same fields. Unlike type 0, type 2 does not change the radio parameters.

Upon the server network receiving a rejoin request of type 0 or 2, the request is sent to the hNS and forwarded to the JS. This server checks the authenticity of the request (verifying MIC value) and generates an incremental number JoinNonce. After that, the server creates a join-accept message and sends it to the hNS. The JS uses RJCount0 instead of DevNonce to calculate all the session security keys.

A rejoin request of type 1 is used to restore lost session context of the server side. This request is sent by ED autonomously or periodically. It contains MHDR, Rejoin Type = 1, JoinEUI, DevEUI, RJcount1, and MIC. Upon this request being received by the NS (hNS or sNS), it is routed to the join server (JS). This server checks the authenticity of the request and generates an incremental number, JoinNonce. After that, the server creates a join-accept message and sends it to the sNS. The JS uses RJCount1 instead of DevNonce to calculate all the session security keys. The counter RJcount1 is incremented with every rejoin request of type 1. The JS keeps track of the last RJcount1 value. This prevents replaying previous type-1 rejoin requests.

Upon the ED receiving a join-accept message, it decrypts it, verifies its authenticity using the MIC, and finally checks whether the JoinNonce value of the message is greater than the last one recorded. The checking of the validity of JoinNonce is a mechanism preventing replaying attacks. If all of these checks succeed, the ED must send an authenticated and encrypted dataUp message with the new session security keys carrying the RekyInd MAC command. The network server (hNS or sNS) validates this message, drops the old session context, and sends a dataDown message carrying the RekyConf MAC command (Figure 9). After that, the ED and the network server can use the new security session.

#### 4.2.3. Security Session Context Generation (ABP Activation)

The ED must be personalized before starting the communication with the network. All the security session keys (AppSKey, FNwkSIntKey, SNwkKeySInt, and NwkSEncKey) and DevAddr are hard-coded into the device. Unlike LoRaWAN v1.0, the frame counters must not be initialized to zero after each reset. An ABP device can be served by a visited network using a passive roaming mode. This server is called the forwarding server network (fNS) (see Figure 10). In this activation mode, the join server is not useful. The NS must store the AppEUI for this end-device by an out-of-band mechanism.

In LoRaWAN v1.1, an ABP-device must send in all uplink messages a ResetInd MAC Command in the FOpts field when it wants to access the NS for the first time or after re-initialization. It stops sending this MAC command when receiving a ResetConf MAC command from the NS. This mechanism is used to inform the NS that the MAC layer context is lost.

#### 4.2.4. Confidentiality of Messages

As in LoRaWAN v1.0.x, join-request/rejoin-request messages are not encrypted.

In a join-accept message, the fields encrypted by JS are JoinNonce, Home_NetID (identifier of hNS), DevAddr, DLSettings, RxDelay, CFList, and MIC. In LoRaWAN v1.1, the key used for the encryption depends on the join-request or rejoin-request message that triggered the sending of the join-accept message. The NwkKey is used if the trigger is join-accept. The JSEncKey is used if the trigger is rejoin-request. As LoRaWAN v1.0, aes128_decrypt is the function used for encryption as follows:(5)$$\begin{array}{c} \mathrm{aes}128\_\mathrm{decrypt}(\mathrm{NwkKey}\mathrm{or}\mathrm{JSEncKey},\;\mathrm{JoinNonce}|\mathrm{Home\_NetID}|\mathrm{DevAddr}|\mathrm{DLSettings}|\mathrm{RxDelay}|\mathrm{CFList}|\mathrm{MIC}) \end{array}$$

All the fields in a dataUp or dataDown are sent in clear except FOpts and FRMPayload fields. The FOpts field carries MAC commands and therefore, the NwkSEncKey is used for encryption. The FRMPayload field can carry a data application or MAC command. In the case of carrying application data, the AppSKey is used for encryption. Otherwise, the NwkSEncKey is used. The encryption of these fields (FOpts or FRMpayload) is done using AES in a counter mode (CTR). The following steps describe how encryption is done for FOptField:aA single block A (16 bytes wide) mainly contains Dir (Dir is 0x00 for uplink frames; otherwise 0x01), DevAddr, and a frame counter (FCntUp for dataUp, NFCntDwn for dataDown carrying MAC Commands, or AFCntDwn for dataDown carrying data application).bCalculating block S by encryption block A using the AES algorithm. The encryption key used is NwkSEncKey generated in the activation phase.cEncryption or decryption is calculated by truncating (FOpts$|{\mathrm{pad}}_{16}$) XOR S to the first “FOptsLen” bytes.

The encryption of the FRMPayload field follows these steps:aDefining a sequence of blocks $A_{i}$/i = 1 ceil(len(FRMPayload)/16) mainly containing Dir, DevAddr, frame counter (FCntUp, NFCntDown, or AFCntDown), and i. Each block has a size of 16 bytes.bCalculating a stream of keys (a sequence S of blocks $S_{i}$) by encryption a sequence of blocks $A_{i}$ using AES algorithm:
S = $S_{1}|S_{i}|\ldots |S_{k}$$S_{i}$ = aes128_encrypt(K,$A_{i}$) i = 1 kK = NwkSEncKey or AppSkeycEncryption or decryption is calculated by truncating (FRMPayload$|{\mathrm{pad}}_{16}$) XOR S to the first len(FRMPayload) bytes.

#### 4.2.5. Integrity and Authenticity of Messages

Like LoRaWAN v1.0, the MIC field is used to authenticate all the messages exchanged between the NS (hNS, sNS, fNS, and JS) and the ED. However, the choice of keys for calculating the MIC is slightly different.

NwkKey is the security key used to sign a join-request by calculating the MIC field. It is calculated as follows:cmac = aes128_cmac(NwkKey, MHDR∣JoinEUI∣DevEUI∣DevNonce)MIC = cmac[0…3]

Since the NwkKey is shared between ED and JS, the integrity of a join-request can only be checked by JS.

The MIC field for a rejoin-request message of type 0 or 2 is calculated using SNwkSIntKey as follows:cmac = aes128_cmac(SNwkSIntKey, MHDR∣Rejoin Type= 0 or 2∣NetID∣DevEUI∣RJCount0)MIC = cmac[0…3]

The use of SNwkSIntKey guarantees that the fNS cannot modify the content of a message of one of these types if it is responsible for forwarding it to the ED.

The MIC field for a rejoin-request message of type 1 is calculated as follows:cmac = aes128_cmac(JSIntKey, MHDR∣Rejoin Type = 1∣JoinEUI ∣DevEUI∣RJCount1)MIC = cmac[0…3]

The use of JSIntKey guarantees that all the network servers that forward this message are not able to modify it. Only the JS can check its integrity.

The MIC field for a join-accept message is calculated as follows:cmac=aes128_cmac(JSIntKey, JoinReqType∣JoinEUI∣DevNonce∣MHDR∣JoinNonce∣NetID∣DevAdd∣DLSettings∣RxDelay∣CFList)MIC=cmac[0…3]

The JoinReqType value indicates the trigger of the join-accept message. The trigger can be a join-request or rejoin-request of type 0, 1, or 2.

The use of JSIntKey guarantees that all the network servers that forward this message are not able to modify it. Only the ED can check its integrity.

For all dataDown and dataUp messages, the MIC is calculated for these fields: MHDR, FHDR, FPort, and FRMPayload. Below, msg designates all the aforementioned fields.

The MIC for dataDown is calculated following these steps:Defining block $B_{0}$ which is mainly composed of ConfFCnt, Dir, DevAddr, AFCntDown or NFCntDown, and len(msg).cmac = aes128_cmac(SNwkSIntKey, $B_{0}|$ msg).MIC=cmac[0…3]

The use of SNwkSIntKey guarantees that the fNS is not able to modify the dataDown message when it forwards it to the ED. The ED can check the integrity of this message.

The MIC for dataUp is calculated following these steps:Defining block $B_{0}$, which is mainly composed of Dir, DevAddr, FCntUp, and len(msg).Defining block $B_{1}$, which is mainly composed of ConfFCnt, TxDr (data rate for uplink transmission), TxCh (index of the channel used for transmission), Dir, DevAddr, FCntUp, and len(msg).cmacS = aes128_cmac(SNwkSIntKey, $B_{1}|$ msg)cmacF = aes128_cmac(FNwkSIntKey, $B_{0}|$ msg)MIC = cmacS[0…1]∣cmacF[0…1]

The MIC field is calculated by concatenating cmacS and cmacF. To check the validity of cmacF, you must have the FNwkSIntKey. This key can be delivered to hNS, sNS and fNS. To check cmacS, you must have the SNwkSIntKey. This key is delivered to hNS and sNS. Thus, the fNS can verify the integrity of the message without modifying it anyway.

In order to acknowledge a dataDown/Up confirmed frame, the sender sends a frame with ACK bit set to 1. In this case, the ConfFCnt field contains the frame counter value modulo $2^{16}$ of the confirmed message; otherwise, ConfFCnt is 0x0000. Thus, the ConfFCnt field indicates which message the sender wants to acknowledge. This mechanism prevents replaying previously cached data-acknowledgment messages.

### 4.3. Third Scenario: ED v1.1 with NS v1.0

In order to make communication possible, ED v1.1 must fall back on v1.0 behavior. This fall-back requires the use of only two counters instead of three on the ED side. Additionally, one rootkey must be used, and only two session security keys must be generated. This rootKey is called NwkKey on the ED side and called AppKey on the NS side.

#### 4.3.1. Message Structure

The message structure has to be the same as described in Section 2.6.1 (first scenario). The ED must not use the rejoin-request message because it is not recognized on the NS side.

#### 4.3.2. Security Session Context Generation (ABP Activation)

According to LoRaWAN v1.1, an ABP-ED shall send in all dataUp messages a ResetInd MAC command when it wants to access the NS for the first time or after re-initializing until receiving a ResetConf MAC command. These two commands are not known by the NS implementing LoRaWAN v1.0.x. LoRaWAN v1.1 does not mention what to do in a compatibility scenario. Additionally, it does not mention how to personalize the ED in a compatibility scenario.

#### 4.3.3. Security Session Context Generation (OTA Activation)

As in the second scenario, the ED sends a join-request carrying JoinEUI, DevEUI, and DevNonce (a sequential number) (see Figure 11). In this scenario, the JS is not used, since the NS implements LoRaWAN v1.0.x. On the network server side, the JoinEUI field of the join-accept message refers to AppEUI. If the ED has permission to join a network, the NS creates a join-accept message and sends it. The structure of this message is the same as the structure described in the first scenario (see Figure 3). Since the NS generates the AppNonce randomly, it is impossible to perform the replay protection mechanism for join-accept like in the second scenario. On the ED side, the JoinNonce field of the join-accept message refers to the AppNonce. In the join-accept message, the first bit of the DLSettings “OptNeg field” is set to zero. In this way, the ED knows that the NS implements LoRaWAN v1.0.x, and the session security keys are calculated as follows:FNwkSIntKey = aes128_encrypt(Nwkkey, 0x01∣AppNonce∣NetID∣DevNonce $|{\mathrm{pad}}_{16}$)SNwkSIntKey = NwkSEncKey = FNwkSIntKeyAppSKey = aes128_encrypt(NwkKey, 0x02∣AppNonce∣NetID∣DevNonce $|{\mathrm{pad}}_{16}$)

On the NS side, the AppKey is the root key used to derive all session security keys which have the same value as the NwkKey used on the ED side. In this scenario, RekeyInd and RekeyConf MAC commands are not used, since these commands are not known by the NS.

In this scenario, the ED and the NS should only use two counters. On the NS side, the FCntUp and FCntDown counters are used, respectively, for dataUp and dataDown. On the ED side, FCntUp is used for dataUp. As for the counter used for dataDown, it can be either AFCntDwn or NFCntDwn. LoRaWAN Specifications do not indicate which counter to use.

#### 4.3.4. Confidentiality of Messages

The join-request message is not encrypted as in the first and the second scenarios.

The join-accept message is encrypted using AppKey on the NS side and decrypted using NwkKey on the ED side. The AppKey and NwkKey have the same value (see Figure 11).

In order to encrypt or decrypt dataUp/Down messages, the ED uses NwkSEncKey when the FRMPayload field carries MAC commands and uses AppSkey when it carries data. On the NS side, NwkSKey is used to encrypt or decrypt MAC commands and AppSKey is used for data applications. Encryption or decryption follows the same steps as described in the first scenario (see Figure 11). The NwkSEncKey and NwkSKey have the same value and are derived from the same RootKey.

In LoRaWAN v1.1, the FOpts field is encrypted, which is not the case for LoRaWAN v1.0. In the specification, there is not any indication related to what the ED must do to be compatible with an NS implementing LoRaWAN v1.0.x. Using the FOpts field to carry encrypted MAC commands does not allow these commands to be interpreted correctly on the NS side, since NS implements LoRaWAN v1.0x. The safest thing to do is to carry all MAC commands in the FRMPayload fields. This was not stated in LoRaWAN v1.1 [23].

#### 4.3.5. Integrity and Authenticity of Messages

Suppose X designates all the fields of the message to be sent.

For a join-request/join-accept messages, the MIC is calculated as follows:cmac = aes128_cmac(RootKey,X)MIC = cmac[0…3]

RootKey designates NwkKey on the ED side. It designates AppKey on the NS Side. We recall that NwkKey and AppKey have the same value.

For a dataUp/Down message, the MIC is calculated as follows:on the NS side
-Defining block $B_{0}$ which is mainly composed of Dir (0: uplink, 1: downlink), DevAddr, FCntUp or FCntDown, and len(X).-cmac = aes128_cmac(NwkSKey, $B_{0}|$ X)-MIC = cmac[0…3]on the ED side
-Defining block $B_{0}$ which is mainly composed of Dir (0: uplink, 1: downlink), DevAddr, FCntUp or FCntDown, and len(X).-cmacF = aes128_cmac(FNwkSIntKey, $B_{0}|$ X)MIC = cmacF[0…3]

Since the ConFCnt field is located in block $B_{1}$, it is not used by the ED. Hence, the mechanism preventing replaying previously cached data-acknowledgment messages is not enabled in this scenario.

### 4.4. Fourth Scenario: ED v1.0 with NS v1.1

In this scenario, the NS v1.1 must fall back on v1.0 behavior. This fall-back requires the use of only two counters instead of three on the backend. The backend is comprised of a network server (NS), Join Server (JS), and an application server (AS), as in scenario 2. In this scenario, the JS has the responsibility of deriving all the session security keys using one root key (AppKey on ED side and NwkKey on NS side), as indicated in [51].

According to LoRaWAN v1.0.x, the width of frame counters can be either 16 bits or 32 bits. LoRaWAN v1.1 uses only frame counters with 32 bits. In this scenario, ED v1.0 with 16-bit-wide frame counters is not compatible with NS v1.1. The specification does not indicate what to do in this case.

Since the join-request message does not indicate the JoinEUI, the NS must be configured with the appropriate JoinEUI by an out-of-band mechanism. The LoRaWAN v1.1 specification [23] does not address this scenario. In the backend specification [51], there are some guidelines on how the network server works to ensure compatibility with ED v1.0.

#### 4.4.1. Message Structure

All messages used in this scenario are the same as used in the first scenario; see Section 2.6.1. The home network server must know the version of the ED by an out-of-band mechanism.

#### 4.4.2. Security Session Context Generation (OTA Activation)

When the NS receives a join-request message from ED, it decides which version of MAC should be used, in v1.0, and transmits its decision to the JS. Thus, JS must derive all security session keys as described in scenario one. In this scenario, the root key is AppKey. The JoinNonce, a sequential value generated by the JS is considered as an AppNonce on the ED side. Since the ED v1.0 considers that AppNonce is generated randomly, the replay protection mechanism for join-accept (newly used in LoRaWAN 1.1) is disabled in this scenario. LoRaWAN v1.1 does not indicate if an NS v1.1 should store a list of last DevNonce as the NS v1.0 to prevent replaying join-request messages. An NS v1.1 is unable to use the RekyInd/RekyConf MAC commands. Additionally, an NS v1.1 is unable to relay for an ED v1.0, since it does not recognize the ForceRejoinReq MAC command.

#### 4.4.3. Security Session Context Generation (ABP Activation)

Before the communication with the network starts, all the security session keys (AppSKey and NwkSKey) and DevAddr are hard-coded into the device. Since the ED v1.0 initializes its frame counters to zero after each reset, the NS 1.1 must do the same thing. This is not indicated in LoRaWAN Specification v1.1. Now the ED v1.0 and NS v1.1 are able to communicate.

In this scenario and with ABP mode, it is impossible to use the REkyInd/RekeyConf MAC commands as described in the second scenario (see Section 4.2.3).

In this activation mode, The NS must store the AppEUI for this end-device with an out-of-band mechanism.

#### 4.4.4. Confidentiality of Messages

The encryption/decryption of all messages exchanged between ED v1.0 and NS v1.1 is similar to that described in scenario one (see Section 4.1.3). In this scenario, the NS v1.1 should not encrypt/decrypt the FOpts fields, since the ED v1.0 does not implement the encryption and decryption algorithm for this field, as specified in LoRaWAN vs. 1.0.

#### 4.4.5. Integrity and Authenticity of Messages

The calculation and the verification of the MIC field are performed as described in the first scenario (see Section 4.1.4).

### 4.5. Synthesis

In Table 2 and Table 3, we summarize the different scenarios already described above. For each scenario, we show the RootKeys, LifteTime Keys, Identifiers, Nonces, and session context (device address, counters for frames, security session Keys), and for each message, which key used to cipher or sign.

## 5. Current Research Directions in LoRaWAN

LoRaWAN is a very promising technology in the field of IoT due to its autonomous network architecture and an open standard specifications. Nevertheless, it needs many enhancements to improve its performance further. We present in this section some open challenges of the LoRaWAN technology.

### 5.1. Network Scalability

Scalability in LoRaWAN is challenging due to the large number of ACKs requested by EDs, the duty cycle restriction imposed in the sub-GHZ ISM band, star topology, and single-hop transmission. In addition, the limited features of LoRaWAN, such as spreading factors (SF) from 7 to 9, bandwidths (125, 250 and 500 kHZ), and code rate (4/5, 4/6, 4/7 and 4/8), prevent the massive deployment of EDs. SF assignment mechanisms, efficient interference cancellation schemes, and novel densification techniques, multi-hop transmission may be solutions to the scalability challenge [52].

### 5.2. Power Consumption

Since nodes in LoRaWAN networks are battery-powered, power consumption is one of the main considerations for evaluating network performance [53]. It should be noted that several works have looked into the power efficiency [54,55,56]. Most of these studies focused on class A, neglecting class B and class C devices. Additionally, energy harvesting is feasible in LoRaWAN according to the results in [57]. This represents an important area of research for LoRaWAN technology.

### 5.3. Security

The number of vulnerabilities and the risk rate associated with a given technology hinder its adoption. This is why security is a major concern for designers of communication protocols. The LoRaWAN Specifications have not forgotten the security issues, despite the limitations of device resources. Indeed, the LoRaWAN Specifications propose several mechanisms to ensure data security, as already explained in Section 4. Several works have been conducted to assess and detect vulnerabilities in LoRaWAN [28,29,30,31,32,33]. Several vulnerabilities have been detected in the different versions of LoRaWAN. To stand out from other LPWAN technologies, LoRaWAN needs to have a high level of security. Hence, researchers have to conduct a lot of research on this issue.

## 6. Conclusions

This article presents a comprehensive study on the security aspects and features of LoRaWAN architectures v1.0.x and v1.1. We detailed four scenarios to study how the NS and ED components in the two architectures interoperate with homogeneous and heterogeneous equipment settings. In particular, we described how EDs and NSs implementing LoRaWAN v1.0 generate session security keys and exchange messages for v1.0, and also described how ED v1.1 and NS v1.1 communicate following the generation of security session keys. We have also designed two new scenarios interchanging the ED and NS versions. For each scenario, we highlighted compatibility issues between the components when implementing the two different LoRaWAN Specifications; and explained how security session context is generated, how to ensure confidentiality of messages, and how the integrity and authenticity of messages are guaranteed in LoRaWAN. Furthermore, we presented various protection schemes used by LoRaWAN protocol against replaying old messages. This work was limited to the LoRaWAN Specifications for devices of class A and did not include an in-depth study of MAC commands. In addition, exchanging messages when roaming was outside the scope of this research. In future, this work could be further extended, such as through studies of class B and C devices in LoRaWAN. We plan to investigate the MAC commands needed to understand how servers can manage their EDs. Moreover, a comprehensive overview of roaming for flexible mobile communication will be prepared.

## Figures and Tables

**Figure 1 sensors-22-03717-f001:**
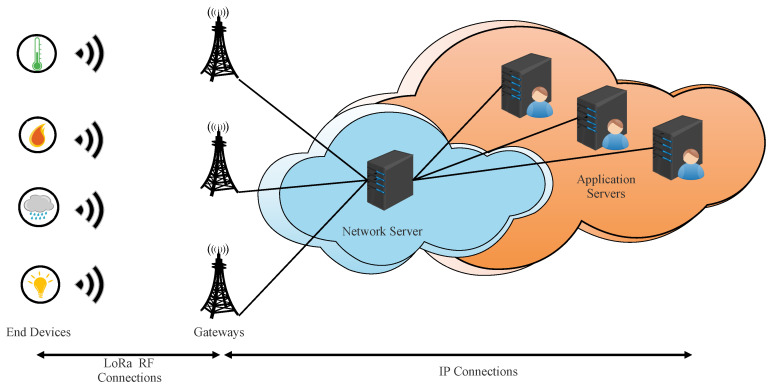
LoRaWAN v1.0 architecture.

**Figure 2 sensors-22-03717-f002:**
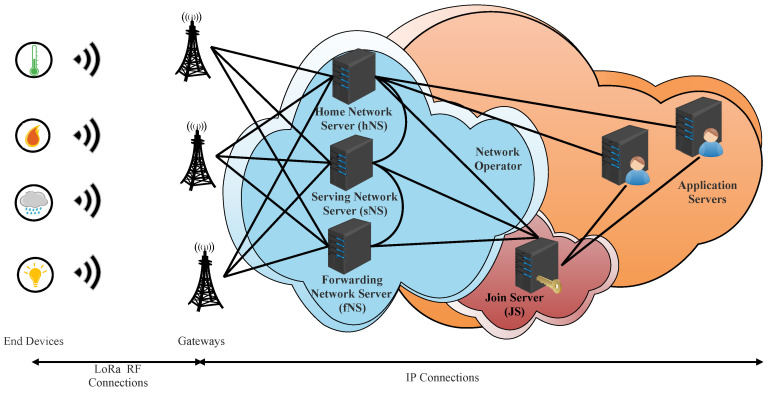
LoRaWAN v1.1 architecture.

**Figure 3 sensors-22-03717-f003:**
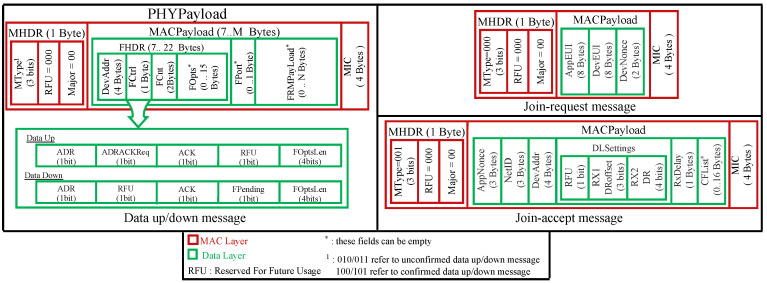
Message structures for LoRaWAN v1.0.x (MAC layer).

**Figure 4 sensors-22-03717-f004:**
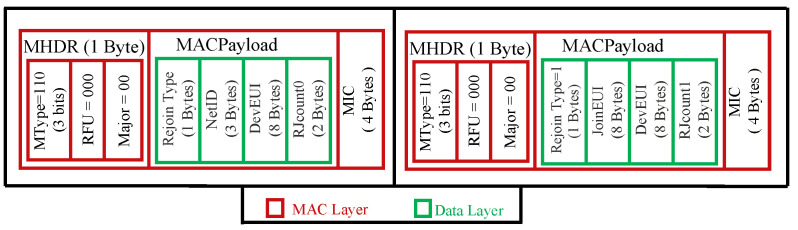
Structures of rejoin-requests types 0, 1, and 2 in LoRaWAN 1.1.

**Figure 5 sensors-22-03717-f005:**
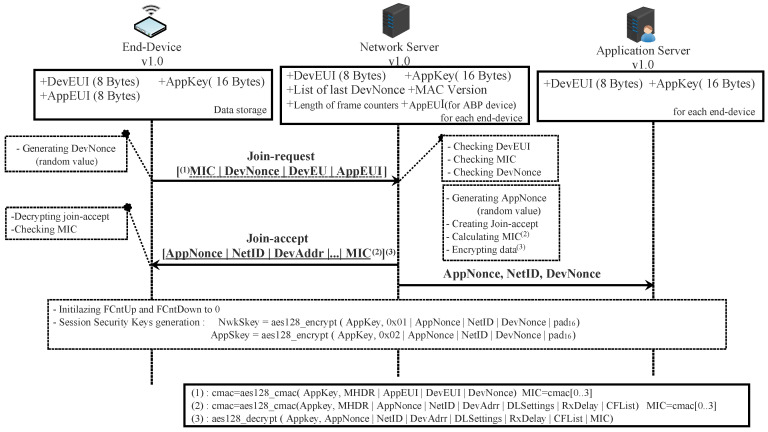
OTA end-device joining a network server in scenario 1.

**Figure 6 sensors-22-03717-f006:**
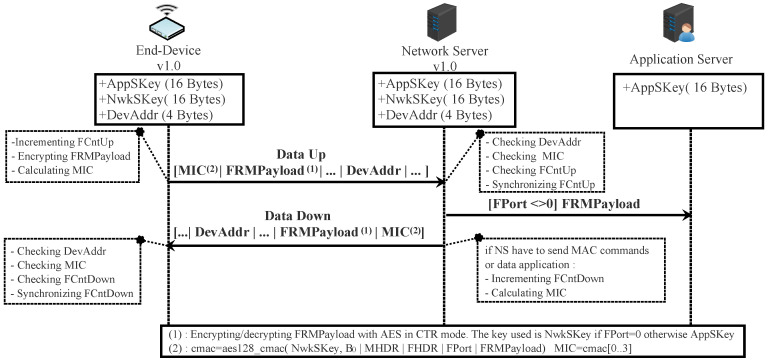
OTA or ABP end-device exchanging messages with a network server in scenario 1.

**Figure 7 sensors-22-03717-f007:**
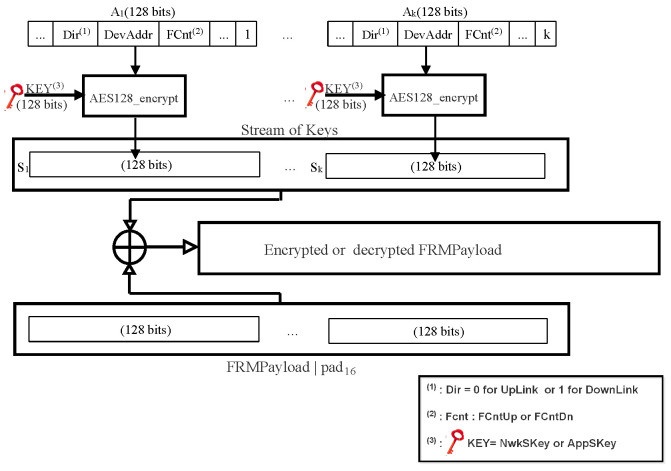
AES 128 in CTR mode.

**Figure 8 sensors-22-03717-f008:**
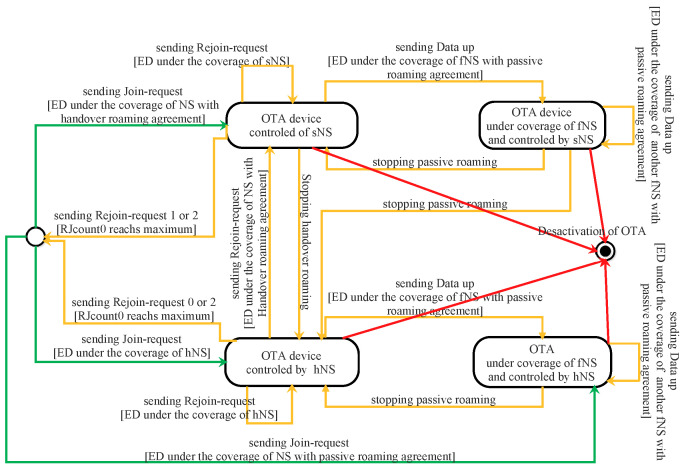
States of an OTA device.

**Figure 9 sensors-22-03717-f009:**
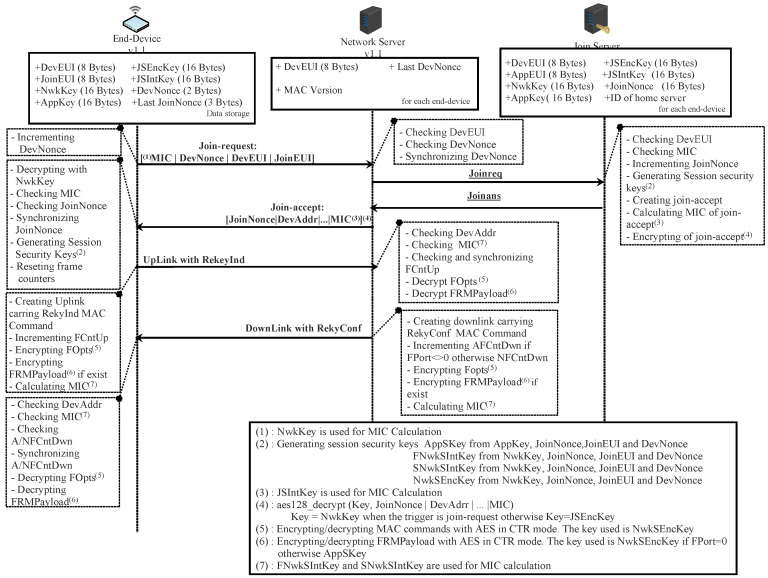
OTA-ED v1.1 joining NS v1.1 in scenario 2.

**Figure 10 sensors-22-03717-f010:**
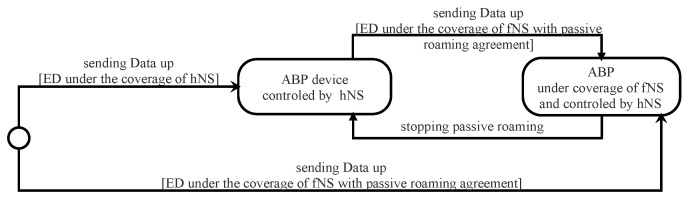
States of an ABP device.

**Figure 11 sensors-22-03717-f011:**
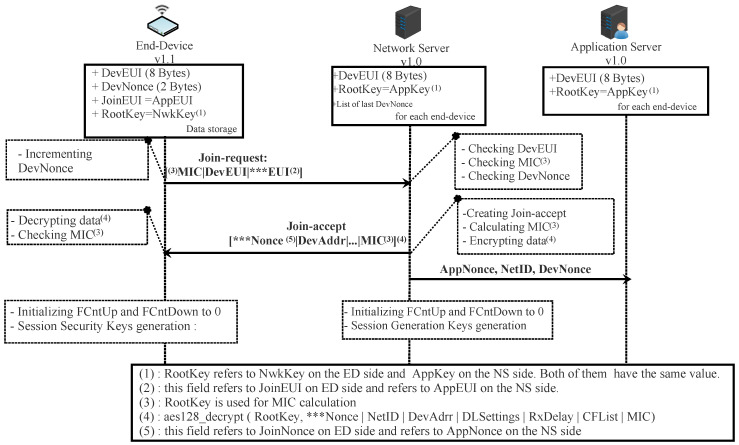
OTA-ED v1.1 joining NS v1.0 in scenario 3.

**Table 1 sensors-22-03717-t001:** Frequency bands.

Frequency Band	Common Name
863–870 MHz	EU868
902–928 MHz	US915
779–787 MHz	CN779
433 MHz	EU433
915–928 MHZ	AU915
470–510 MHz	CN470
923 MHz	AS923
920–923 MHz	KR920
865–867 MHZ	IN865
864–870 MHZ	RU864

**Table 2 sensors-22-03717-t002:** A comparison between scenarios 1 and 2.

	**Scenario 1**		**Scenario 2**	
**Root Keys** **Lifetime Keys**	AppKey - -		AppKey NwkKey JSEncKey (from NwkKey) JSIntKey (from NwkKey)	
**Identifiers**	DevEUI - AppEUI		DevEUI JoinEUI AppEUI (storage on NS)	
**Nonce used for deriving** **security session**	DevNonce (randomly) AppNonce (randomly)		DevNonce (Incremental) JoinNonce (Incremental) RJcount0 (Incremental) RJcount1(Incremenatl)	
**Session context** **-address of device** **-frame counters** **-security session keys**	DevAddr FCntUp FCntDown AppSKey (from AppKey) NwkSKey (from AppKey)		DevAddr FCntUp NFCntDwn AFCntDwn AppSKey (from AppKey) NwkSEncKey (from NwkKey) SNwkSIntKey (from NwkKey) FNwkSIntKey (from NwkKey)	
**Exchanged messages**	**Ciphered by**	**Signed by**	**Ciphered by**	**Signed by**
**-Join-request** **-Rejoin-request type 0, 2** **-Rejoin-request type 1** **-Join-accept (reply to Join-request)** **-Join-accept (reply to Rejoin-request)** **-Data Up** **FRMPayload (MAC Commands)** **FRMPayload (Data)** **FOpts** **-Data Down** **FRMPayload (MAC Commands)** **FRMPayload (Data)** **FOpts**	Plain Text Not used Not used AppKey Not used NwkSKey AppSkey Plain Text NwkSKey AppSkey Plain Text	AppKey Not used Not used AppKey Not used NwkSKey NwkSkey	Plain Text Plain Text Plain Text NwkKey JSEncKey NwkSEncKey AppSKey NwkSEncKey NwkSEncKey AppSkey NwkSEncKey	NwkKey SNwkSIntKey JSIntKey JSIntKey JSIntKey SNwkSIntKey + FNwkSIntKey SNwkSIntKey

**Table 3 sensors-22-03717-t003:** A comparison between between scenarios 3 and 4.

	**Scenario 3**				**Scenario 4**			
	**ED v1.1 Side**		**NS v1.0 Side**		**ED v1.0 Side**		**NS v1.1 Side**	
**Root Keys** **Lifetime Keys**	NwkKey -		AppKey -		AppKey -		NwkKey -	
**Identifiers**	DevEUI JoinEUI		DevEUI AppEUI		DevEUI AppEUI		DevEUI JoinEUI (out-of-band mechanism)	
**Nonce used for deriving** **security session**	DevNonce (Incremental) JoinNonce(keep track)		DevNonce (keep track) appNonce(randomly)		DevNonce (randomly) AppNonce(No track)		DevNonce (Keep track) JoinNonce(Incremental)	
**Session context** **-address of device** **-frame counters** **-security session keys**	DevAddr FCntUp FCntDown AppSkey(from NwkKey)		DevAddr FCntUp FCntDown AppSKey(from AppKey)		DevAddr FCntUp FCntDown AppSKey(from AppKey)		DevAddr FCntUp FCntDown AppSKey(from NwkKey)	
	FNwkSIntKey(from NwkKey) = SNwkSIntKey = NwkSEncKey		NwkSKey(from AppKey)		NwkSKey(from AppKey)		NwkSKey(from NwkKey)	
**Exchanged messages**	**Ciphered by**	**Signed by**	**Ciphered by**	**Signed by**	**Ciphered by**	**Signed by**	**Ciphered by**	**Signed by**
**-Join-Request (JR)** **-ReJoin-Request(RJR) type 0, 2** **-ReJoin-Request(RJR) type 1** **-Join-accept (reply to JR)** **-Join-accept (reply to RJR)** **-Data Up** **FRMPayload (MAC Commands)** **FRMPayload (Data)** **FOpts** **-Data Down** **FRMPayload (MAC Commands)** **FRMPayload (Data)** **FOpts**	Plain Text Not used Not used NwkKey Not used - NwkSEncKey AppSKey Plain Text - NwkSEncKey AppSkey Plain Text	NwkKey Not used Not used NwkKey Not used FNwkSIntKey SNwkSIntKey	Plain Text Not used Not used AppKey Not used - NwkSKey AppSKey Plain Text NwkSKey AppSkey PlainText	AppKey Not used Not used AppKey NwkSkey NwkSKey	Plain Text Not used Not used AppKey Not used - NwkSKey AppSKey Plain Text - NwkSKey AppSkey Plain Text	AppKey Not used Not used AppKey Not used NwkSKey NwkSKey	Plain text Not used Not used NwkKey Not used - NwkSEncKey AppSKey Plain Text - NwkSEncKey AppSkey Plain Text	NwkKey Not used Not used NwkKey Not used NwkSKey NwkSKey

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
