# Peer review of "Analysis of LoRaWAN 1.0 and 1.1 Protocols Security Mechanisms"

_sensors, 2022, doi:10.3390/s22103717_

Round 1

Reviewer 1 Report

The present work entitled "Analysis of LoRaWAN 1.0 and 1.1 Protocols Security Mechanisms" shows 4 different scenarios of use of end devices and network servers crossing the different LoraWAN protocols (V1.0 and V1.1). The article is quite consistent, however you should take into account the following:
1. The statement in the abstract "... and now is the mainstream communication protocol for Internet of Things technologies" is supported in which study?
2. Line 95, do the references change from 11 to 38? They need to be re-indexed.
3. In line 97 "LoRa modulation technology uses the ISM band (Industrial, Scientific, and Medical). It is an unlicensed band", the frequencies must be specified, as these are different in different regions of the world.
4. Line 107 to line 123, must be supported with references
5. Figure 1 and 2 require connections between End Devices and Gateways.

6. line 891 points to figure 4.1.3? and line 896 for figure 4.1.4?
7. The conclusions and future work are very limited in terms of explaining in detail what was expected and in accordance with the objectives and should have a focus on results and the way in which objectives were achieved.

Author Response

The present work entitled "Analysis of LoRaWAN 1.0 and 1.1 Protocols Security Mechanisms" shows 4 different scenarios of use of end devices and network servers crossing the different LoraWAN protocols (V1.0 and V1.1). The article is quite consistent, however you should take into account the following: 
1. The statement in the abstract "... and now is the mainstream communication protocol for Internet of Things technologies" is supported in which study? 
2. Line 95, do the references change from 11 to 38? They need to be re-indexed. 
3. In line 97 "LoRa modulation technology uses the ISM band (Industrial, Scientific, and Medical). It is an unlicensed band", the frequencies must be specified, as these are different in different regions of the world. 
4. Line 107 to line 123, must be supported with references 
5. Figure 1 and 2 require connections between End Devices and Gateways.
6. line 891 points to figure 4.1.3? and line 896 for figure 4.1.4? 
7. The conclusions and future work are very limited in terms of explaining in detail what was expected and in accordance with the objectives and should have a focus on results and the way in which objectives were achieved.

Author response: 

Thanks, reviewer.  We agree with the reviewer’s comment. In an effort to improve the overall flow : 

  • We changed the statement "... and now is the mainstream communication protocol for Internet of Things technologies" in the abstract section
  • All references have been re-indexed.
  • In subsection "2.2. LoRa Technology", we specified for the ISM band the frequencies used in different regions of the world.
  • In subsection "2.3. LoRaWAN Protocol", we added references for each LoRaWAN specification.
  • Connections between ED and GW have been added in Figures 1 and 2.
  • In sections "4.4.4 Confidentiality of Messages" And "4.4.5. Integrity and Authenticity of Messages" the word "Figure" have been replaced by "Section"
  • Section “5. Conclusion” has been revised taking into account the reviewer's comment.

Reviewer 2 Report

Dear Authors

Thank you very much for addressing my comment.

The manuscript looks OK now.

good luck! 

Author Response

Thanks, reviewer.

Reviewer 3 Report

The article reviews LoRaWAN as one of the most commonly used Internet of Things (IoT) networks. The main contributions of the work are presented; however, I have the following concerns.

  • The novelty of the work is poor. Most of the mentioned data are available in the technical papers of LoRaWAN.
  • The section of related works needs to be reconsidered. Most of the considered works are far from the scope of this article. They may be considered in a separate section as applications of LoRaWAN. Existing proposals in the context of LoRaWAN should only be introduced and compared to the existing work.
  • Limitations of the considered LoRaWAN protocols should be introduced.
  • Distributed edge computing, including fog computing, is a main paradigm introduced recently to the LoRaWAN, and it has facilitated the implementation of security algorithms. Thus, it should be introduced and the existing works that considered security issues of MEC/fog-based LoRaWAN should be considered.
  • Current research directions in LoRaWAN should be introduced in a separate section.

Author Response

Thanks, reviewer.  We agree with the reviewer’s comment. In an effort to improve the overall flow : 

  • The "Related Works" section has been revised taking into account the reviewer's comment. Thus, many works considered in this section have been eliminated and the others are within the scope of the article.
  • In the section "3. Related Works", we added some works dealing with limitations of LoRaWAN protocol

In addition, we have added some works on Fog computing and issues related to its security requirements.

Round 2

Reviewer 1 Report

Good work.

All questions have been addressed and explained.

Author Response

Thanks, reviewer.

Reviewer 3 Report

The authors have not addressed these comments yet.

  • Limitations of the considered LoRaWAN protocols should be introduced.
  • Distributed edge computing, including fog computing, is a main paradigm introduced recently to the LoRaWAN, and it has facilitated the implementation of security algorithms. Thus, it should be introduced and the existing works that considered security issues of MEC/fog-based LoRaWAN should be considered.
  • Current research directions in LoRaWAN should be introduced in a separate section

Author Response

Thanks, reviewer.

We agree with the reviewer’s comments. In an effort to improve the overall flow, we:

Point 1: added a "LoRaWAN Limitations" subsection below the "LoRaWAN" section (from line 143).

Point 2:  added a paragraph in the "Introduction" section to introduce the paradigm of fog computing in LoRaWAN (from line 54).

Point 3: added a separate section “Current Research Directions in LoRaWAN" (from line 875).

Round 3

Reviewer 3 Report

All comments have been addressed.